# Effects of Acculturation Types on Acculturative Stress and Adjustment to South Korean Society: Focusing on Chinese Immigrants

**Bo Eun Jung**

Center for International Area Studies, Hankuk University of Foreign Studies, Seoul 02450, Korea;
qingtian88@naver.com or qingtian88@hufs.ac.kr

**Abstract:** This study aims to empirically analyze the effects of acculturation types of Chinese immigrants who have settled in South Korea on their acculturative stress and adjustment to South Korean society. For this, 200 Chinese immigrants residing in Korea were surveyed. Of these, 12 were excluded from the survey for insincere or omitted responses, and the final 188 were surveyed. The research results are as follows: First, the effects of the types of acculturation of Chinese immigrants on their acculturative stress were analyzed. According to the results, among the subfactors of acculturation type, integration and assimilation had significant negative effects on acculturative stress, and marginalization had significant positive effects. Second, the effects of immigrants' acculturative stress on their adjustment to South Korean society were analyzed, and it was found that their acculturative stress had significant negative effects on their adjustment to South Korean society. Third, the effects of immigrants' acculturation types on their adjustment to South Korean society were analyzed. Among the subfactors of the acculturation types, integration and assimilation were found to have significant positive effects on adjustment to South Korean society, while marginalization had significant negative effects. Fourth, the mediating effect of acculturative stress on the relationship between the integration of Chinese immigrants and their adjustment to South Korean society was analyzed. As a result, it was found that the integration, separation, and marginalization of immigrants had significant indirect effects on their adjustment to South Korean society through acculturative stress. This study can be regarded as meaningful in that it presented the acculturation types necessary for immigrants, who are steadily increasing in South Korea in this era of globalization, to relieve the acculturative stress they feel in an unfamiliar foreign country and adjust to South Korean society.

**Keywords:** Chinese immigrants; acculturation types; acculturative stress; adjustment to society

## 1. Introduction

As the development of transportation, communication, and scientific technologies has enabled smooth exchanges of information and manpower among cultural areas around the world, political, economic, and social networks among countries are strengthened, and the movement of populations is becoming common in the world [1]. Studying abroad and overseas travel have become easier compared to the past, and so the number of overseas tourists and international students has increased, and as domestic and foreign exchanges of labor force become active due to the increase in the overseas expansion of companies and in the recruitment of foreign workers, immigration is also increasing [2]. As a result of the globalization of countries, Korean society is undergoing many changes economically and culturally, and as we see people with various languages and cultural backgrounds around us, the phenomenon of international migration appears in various forms in our lives [3]. As South Korea gradually became a member of international society, the rise of international migration rapidly occurred. In particular, due to the influx of foreign workers and the increase in international marriages, immigrants with various languages and cultural backgrounds have been spreading [4,5]. In the case of South Korea,

migration through international marriages has accelerated since 2000, and this has led to an increase in marriage immigrants. As the number of multicultural families grew and social interest consequently rose due to increases in immigrants, the South Korean government installed the Multicultural Family Policy Committee in the Prime Minister's Office in 2009 to generalize and coordinate multicultural family support projects. In May 2010, the Prime Minister's Office and related ministries jointly established the Basic Plan for Multicultural Family Support Policies to prepare basic directions for multicultural family support projects from 2010 to 2012. The basic plan was reviewed and confirmed in March 2011 and began to be implemented. Currently, the 3rd Basic Plan for Multicultural Family Support Policies (2018~2022) is being implemented [6]. Since the government policies intended to respond to multicultural society were hurriedly formed under the leadership of the government focusing on multicultural families, which rapidly increased in the 2000s, they could produce good outcomes for the stable adjustment of multicultural families to South Korean culture in a short time, but they emphasized the familial aspect rather than the human rights aspect of immigrants because they focused on stable family lives [7–9]. That is, since those policies focused on programs for the formation of smooth family relationships and role performance at home rather than strengthening the independent capacity of immigrants, they enabled immigrants to stably adjust to South Korean culture centered on families, but they had limitations in enabling immigrants to participate in social activities as independent members of society in the long term [10,11]. In addition, the practicality and effectiveness of those acculturation policies were insufficient in the field because there were some differences in perception between the policies of the central government and local governments and immigrants, although acculturation policies should be implemented in linkages among the three [12].

The concept of acculturation was introduced in 1936 by Redfield, Linton & Herskovits [13]. Acculturation comprehensively refers to phenomena resulting from the continuous contact of immigrant groups with culturally different backgrounds from the culture of indigenous groups and is a concept that includes continuous changes that occur in the culture of the indigenous group as well as in the culture of the immigrant groups [14–17]. Immigrants can feel a great burden due to the sense of loss that comes from the process of acculturation in the country where they settle, and this can appear as stress. This acculturative stress is a concept that can represent the psychological well-being arising from personal relationships and psychological satisfaction [18–20]. Structural chaos, cultural clashes, and cultural alienation in the process of cultural changes can also cause acculturative stress, and such stress should be dealt with at the social level because it makes social integration difficult and causes a great crisis for immigrants [21–23].

According to Statistics Korea data [24], the total number of multicultural households in Korea is 346,017, with 82.4% of marriage immigrants and 17.6% of other naturalized households. A "multicultural society" refers to a society that is ethnically, racially, and culturally diversified. This means that there are many different lifestyles within a country or society. The main reason that Korea has become a multicultural society is the increase in foreign workers and international marriages. As such, the number of multicultural families in Korea is steadily increasing every year, and there is a transition from a mono-ethnic culture to a multicultural society. According to data from Statistics Korea, as of May 2021, there were 1332 thousand foreigners residing in Korea over the age of 15 [24]. By nationality, there were 511.4 thousand Korean-Chinese and 1,267,000 Chinese immigrants, representing 638.1 thousand Chinese immigrants, accounting for 47.9% of all Korean immigrants [24]. Chinese immigrants, who make up the majority of immigrants in Korea, have different cultural backgrounds and experience acculturation stress as they settle into Korean society.

In previous studies, it was shown that factors such as difficulties in language communication [25], short residence periods, large age gaps with husbands, domestic violence [26], the husband's low social status, conflicts between family members, social prejudices about international marriages, limited opportunities for economic activities, a patriarchal social culture, difficulties in interpersonal relationships [27,28], religious and cultural differences,

and economic difficulties [29] negatively affect the acculturation process of Chinese female marriage immigrants [30].

The changes in cultural situations and conflicts experienced by female marriage immigrants at home cause acculturative stress, and when an individual's ability to cope with the stress reaches the limit, they experience psychological difficulties such as confusion, anxiety, depression, loneliness, and alienation [31].

Therefore, the successful acculturation of immigrants can be said to be a very important issue not only for immigrants and their families but also for the future of the entire South Korean society, which is becoming a multicultural society. Whereas successful acculturation can bring about the self-development, psychological well-being, and social development of immigrants, failure to adjust can lead to social falling behind, frustration, and social losses [32]. Therefore, for immigrants to settle stably as independent members of South Korean society, examining the relationship between the acculturative stress of immigrants according to their acculturation types and the level of their adjustment to South Korean society will be meaningful in that it enables seeking policy measures for the smooth acculturation of immigrants from a multiculturalist perspective. Therefore, the purpose of this study is to empirically analyze the effects of the acculturation types of Chinese immigrants who have settled in South Korea on their acculturative stress and adjustment to South Korean society.

## 2. Theoretical Background

### 2.1. Concept and Types of Acculturation

In general, adjustment refers to changes occurring in an organism as a survival response to environmental demands, and appropriate adjustment is required in changes such as a movement from a particular culture or environment to another one [33]. Therefore, acculturation can be seen as all the processes of changes through long-term contact between two groups of people with different cultures in a new cultural and social environment [22,34]. From a cultural point of view, the adjustment patterns of individuals are not simple or uniform but vary according to developmental states and situational contexts [16,35]. Berry (1997) presented four types of acculturation through two criteria [36]. The first criterion is "cultural maintenance", which means the degree to which immigrants perceive the identity of their original culture when they adjust to a new culture. The second criterion is "contact and participation", which indicates the degree to which immigrants make contacts and exchanges with new cultures. According to these criteria, Berry (1997) classified acculturation types into integration, assimilation, separation, and marginalization [37].

First, integration refers to cases where immigrants accept the culture of the new society while maintaining their homeland culture by successfully integrating both cultures, and the immigrants have competent bicultural attitudes. Second, assimilation refers to cases where immigrants abandon their homeland culture and accept only the culture of the new society, and in the process of participating in the new society, they abandon their own culture and identity and are absorbed into the mainstream society group. Third, separation refers to cases where immigrants maintain their homeland culture and refuse the culture of the new society, and they continuously maintain the identity of the culture to which they originally belonged. Fourth, marginalization refers to cases where immigrants can neither maintain their homeland culture nor accept the culture of the new society and stay on the periphery of the mainstream culture. They mainly fall into the lower classes of society and may have values and behavior patterns that rebel against the existing order and culture. In conclusion, acculturation can be viewed as something intended to understand the phenomena that occur through continuous direct contact between two groups of individuals who already have their cultural tendencies and secondarily have different cultures [22].

Factors affecting acculturation include personal characteristic factors and environmental factors. Personal characteristic factors such as gender, age, education, knowledge, and economic level are known to have large effects on the acculturation of immigrants [22,38,39].

That is, those who are younger at the time of immigration and females adjusted better to the new culture [40], and those whose time of immigration was earlier and those with higher levels of education had lower levels of acculturative stress [40,41]. In addition, it was found that immigrants with voluntary migration motives adjusted to the new culture much more easily than those with involuntary migration motives [22,27].

### 2.2. Acculturative Stress

Acculturative stress is an extension of Lazarus (1984)'s concept of stress, and it can be caused by the inability to adjust required of individuals in an unfamiliar culture beyond the internal and external capabilities of the individuals to the extent that they are excessively affected or by their sense of loss of their belief and support system in the other culture [42]. Acculturative stress may be a state of tension caused by the dysfunction experienced by individuals or groups in the process of new acculturation, and they may experience maladjustment due to negative emotions such as depression, anxiety, and withering [43,44]. The definition of acculturative stress differs between researchers. Berry (2020) [22] defined it as the difficulties, psychological conflicts and confusion, psychological pressure, anxiety, tension, etc., experienced during the process of acculturation, and Sahdhu & Asrabadi (1994) [45] defined it as a certain behavioral type due to stress in the process of acculturation and presented its components such as perceived discrimination, hostility, and hatred, homesickness, fear, culture shock, and guilt.

Acculturative stress can explain even social problem behaviors along with mental health problems such as psychological and emotional pains experienced by immigrants through the process of adjustment to an unfamiliar culture. Therefore, since acculturative stress factors have not only common but also different aspects depending on the subject, married immigrant women who have difficulties in various complex areas such as family and economy experience acculturative stress due to family relationships and identity [46]. Acculturative stress includes physical, psychological, and social aspects, and is accompanied by certain stress behaviors such as negative mental health conditions including anxiety or depression, feelings of alienation, physical symptoms, and identity confusion [47,48].

The levels of acculturative stress differed according to the acculturation types classified by Berry (integration, assimilation, separation, and marginalization), and it was argued that the level of acculturative stress was the lowest in the case of integration and the highest in the case of separation [32]. According to Jang et al. (2004) [49], who examined stress levels according to the characteristics of acculturation groups, groups with a strong tendency to voluntarily enter an unfamiliar culture (international students or sojourners) showed lower acculturative stress than involuntary cultural groups (refugees, etc.) and stress levels differed according to socioeconomic statuses or personal characteristics even in groups with a strong tendency to voluntarily enter an unfamiliar culture. That is, the issue of perceived discrimination, a sense of alienation, language problems, the lack of available social and economic resources, and separation from cultural pride can also be factors that increase acculturative stress [50,51].

### 2.3. Adjustment

Adjustment refers to a state in which a person feels satisfied without frustration or anxiety in daily life by achieving harmony between an individual's internal psychological needs and the external social environment [34]. Adjustment related to migration can be divided into socio-cultural adjustment and psychological adjustment. Sociocultural adjustment refers to acquisition of language, cultural understanding, and value judgment in the country of residence, while psychological adjustment refers to psychological satisfaction or happiness in the country of residence. The two are in a relationship of mutual influence. When a migrant is in a state of psychological maladjustment such as loneliness or stress, the migrant's life becomes unstable and social and cultural adjustment is also difficult [52]. For smooth adjustment, it is necessary to consider both.

According to Berry (1997) [37], social adaptation is explained in terms of integration, separation, assimilation, and marginalization. Among the four dimensions of social adaptation, integration refers to a case in which the culture of the local society is accepted while maintaining the cultural identity and characteristics of one's country. Separation refers to maintaining their original ethnic identity and avoiding cross-cultural contact. Assimilation is about severing ties with one's own culture and adapting to the lifestyle of a new culture. Marginalization refers to the characteristics of individuals who have lost their relationship with both the original culture and the new culture. In general, the adaptation of those who use the strategy of integration is excellent, but the adaptation of those who adopt the strategy of marginalization in which they are marginalized both in their own country and in the country of residence with a defensive attitude will have difficulties in adaptation [53].

## 3. Method

### 3.1. Survey Participants

In this study, foreign immigrants in Korea were set as the population, and among them, Chinese immigrants in their 20s and 30s, who show the highest immigration rate, were used as the sample group. The questionnaire was distributed to Chinese students who were active in the Chinese student community in Korea from the 1st week to the 4th week of April 2022. At this time, a Google Docs questionnaire was developed to conduct online surveys and distributed through e-mail or messenger. Respondents who agreed to the purpose of the questionnaire filled out the questionnaire in the form of self-filling.

The general characteristics of the Chinese immigrants who are the survey participants are shown in Table 1. Out of a total of 200 participants, 12 participants who respond insincerely or omitted some responses were excluded and the responses of 188 participants were used in this study. By age, 163 participants (86.7%) were in their 20s and 25 (13.3%) in their 30s, and by level of education, 47 participants (25.0%) were college graduates and 141 participants (75.0%) were at least graduate school graduates or educated more. By residence period, 82 participants (43.6%) resided for less than 3 years in South Korea and 106 participants (56.4%) resided for at least 3 years in South Korea and returned to their homeland, 42 participants (22.3%) and the intention and 146 participants (77.7%) had no such intention.

**Table 1.** General characteristics of survey participants.

| Division | | N | % |
|---|---|---|---|
| Age | 20s | 163 | 86.7 |
| | 30s | 25 | 13.3 |
| Level of education | College graduate | 47 | 25.0 |
| | Graduate school graduates or educated more | 141 | 75.0 |
| Residence period | Less than 3 years | 82 | 43.6 |
| | At least 3 years | 106 | 56.4 |
| Intention to return to the homeland | Yes | 42 | 22.3 |
| | No | 146 | 77.7 |
| Total | | 188 | 100.0 |

### 3.2. Research Model and Hypothesis

#### 3.2.1. Research Model

In this study, variables were designed so that acculturation types of Chinese immigrants living in Korea were used as an independent variable and adjustment to South Korean society was used as a dependent variable, and acculturative stress was input as a parameter to survey mediating effects on the causal relationships between the foregoing variables. The acculturation types of Chinese immigrants were composed of four subfactors, which are integration, assimilation, separation, and marginalization based on the

studies conducted by Berry (2001) [54] and Kim (2018) [31]. Both acculturative stress and adjustment to South Korean society were regarded as single factors. The model of this study is as follows (see Figure 1).

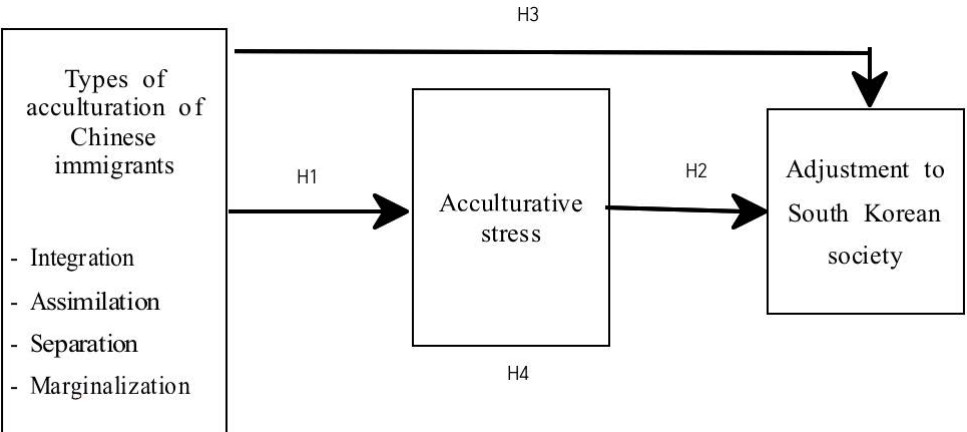

**Figure 1.** Research model.

3.2.2. Research Hypothesis

The following hypotheses were derived based on the preceding studies of this study and the above study model.

Most immigrants who come to Korea experience various difficulties in the process of adapting to a new culture because they lack preparation for Korean culture and do not have enough time to adapt to society. Immigrants show various types of adaptation, such as recognizing the identity of an existing culture as more important than a new culture, or making contact with and interacting with a new culture [55]. According to these types, the acculturation stress experienced by immigrants is different. Lim et al. (2010) [56] examined the relationship between the acculturation type and the acculturation stress of female marriage immigrants and confirmed that the acculturation type is a leading variable affecting acculturation stress. Based on this, the following Hypothesis 1 was established.

**H1.** *The types of acculturation of Chinese immigrants will affect their acculturative stress.*

**H1-1.** *The integration of Chinese immigrants will negatively (−) affect their acculturative stress.*

**H1-2.** *The assimilation of Chinese immigrants will negatively (−) affect their acculturative stress.*

**H1-3.** *The separation of Chinese immigrants will positively (+) affect their acculturative stress.*

**H1-4.** *The marginalization of Chinese immigrants will positively (+) affect their acculturative stress.*

Because acculturation stress can be caused by a sense of loss of one's own beliefs and support systems within an unfamiliar culture [22], it leads to negative mental health conditions such as anxiety or depression or feelings of alienation, which negatively affects social adaptation [47]. Lee et al. (2018) [57] confirmed that the satisfaction level of school life of Chinese students affects acculturation stress, and this stress affects Chinese students' adaptation to Korean society. Based on these results, the following Hypothesis 2 was established.

**H2.** *The acculturative stress of Chinese immigrants will positively (+) affect their adjustment to South Korean society.*

Berry (1997) [36] suggests two criteria for classifying acculturation types. The first criterion is 'cultural maintenance', which recognizes the identity of the existing culture as more important than the new culture. The second criterion is 'contact and participation' to contact and exchange with a new culture. According to these criteria, social adaptation types

can be divided into integration and assimilation, and separation and marginalization [36]. Based on this, the following Hypothesis 3 was established.

**H3.** *The types of acculturation of Chinese immigrants will affect their adjustment to South Korean society.*

**H3-1.** *The integration of Chinese immigrants will negatively (−) affect their adjustment to South Korean society.*

**H3-2.** *The assimilation of Chinese immigrants will negatively (−) affect their adjustment to South Korean society.*

**H3-3.** *The separation of Chinese immigrants will positively (+) affect their adjustment to South Korean society.*

**H3-4.** *The marginalization of Chinese immigrants will positively (+) affect their adjustment to South Korean society.*

Considering that the type of acculturation is a variable influencing acculturation stress [56] and that acculturation stress is a factor influencing adaptation to Korean society [58], it can be predicted that this is a variable mediating the relationship between the type of adaptation and social adaptation. Therefore, the following Hypothesis 4 was established.

**H4.** *Chinese immigrants' acculturative stress will have mediating effects on the relationship between the types of acculturation of Chinese immigrants and their adjustment to South Korean society.*

**H4-1.** *Chinese immigrants' acculturative stress will have mediating effects on the relationship between the integration of Chinese immigrants and their adjustment to South Korean society.*

**H4-2.** *Chinese immigrants' acculturative stress will have mediating effects on the relationship between the assimilation of Chinese immigrants and their adjustment to South Korean society.*

**H4-3.** *Chinese immigrants' acculturative stress will have mediating effects on the relationship between the separation of Chinese immigrants and their adjustment to South Korean society.*

**H4-4.** *Chinese immigrants' acculturative stress will have mediating effects on the relationship between the marginalization of Chinese immigrants and their adjustment to South Korean society.*

*3.3. Reliability of Measurement Tools*

In this study, the item compositions and reliability of the types of acculturation, acculturative stress, and adjustment to South Korean society scales were investigated, and the results are presented in Table 2.

The independent variable, the types of acculturation, consisted of 4 subareas, which comprised a total of 28 items consisting of 5 items for integration, 8 items for assimilation, 6 items for separation, and 9 items for marginalization questions. The parameter acculturative stress comprised 16 questions, and the dependent variable adjustment to South Korean society comprised 11 items. The reliability of the measurement items was analyzed, and based on the results, item no. 14 for separation among the types of acculturation was excluded because it was shown to impair reliability. Consequently, the reliability values of the subareas of the types of acculturation were as follows; integration: Cronbach's $\alpha = 0.822$, assimilation: Cronbach's $\alpha = 0.778$, separation: Cronbach's $\alpha = 0.821$, and marginalization: Cronbach's $\alpha = 0.893$. The reliability of the parameter acculturative stress was Cronbach's $\alpha = 0.893$, and that of the dependent variable adjustment to South Korean society was Cronbach's $\alpha = 0.901$. The Cronbach's $\alpha$ values of all variables were shown to be at least 0.60, indicating very high reliability in general.

**Table 2.** Reliability of measurement tools.

| Variables | | Number of Items | Questionnaire Items | Reliability (Cronbach's) |
|---|---|---|---|---|
| Acculturation type | Integration | 5 | 3, 7, 11, 15, 18 | 0.822 |
| | Assimilation | 8 | 1, 5, 9, 13, 17, 20, 23, 26 | 0.778 |
| | Separation | 5 | 2, 6, 10, 21, 24 | 0.821 |
| | Marginalization | 9 | 4, 8, 12, 16, 19, 22, 25, 27, 28 | 0.893 |
| Acculturative stress | | 16 | 1, 2, 3, 4, 5, 6, 7, 8, 9, 10, 11, 12, 13, 14, 15, 16 | 0.893 |
| Adjustment to South Korean society | | 11 | 1, 2, 3, 4, 5, 6, 7, 8, 9, 10, 11 | 0.901 |

*3.4. Data Processing*

The data collected in this study were processed using the statistical programs SPSS 27.0 and SPSS Macro (Model 4) 3.4, and analyses were conducted as follows. First, frequency analysis was conducted to find out the general characteristics of the survey subjects Chinese immigrants, and the frequency and percentage (%) were calculated. Second, descriptive statistics (average, standard deviation) were analyzed to understand the types of acculturation, acculturative stress, and level of social adjustment of Chinese immigrants in South Korea. Third, reliability analyses (Cronbach's $\alpha$) were conducted to find out the reliability of the measurement tools used to measure the variables in this study, which are the types of acculturation, acculturative stress, and social adjustment in South Korea. Fourth, independent t-tests were conducted to examine the differences in acculturation types, acculturative stress, and social adjustment in South Korea according to the general characteristics (age, education levels, residence period, intention to return to their homeland) of Chinese immigrants, who were survey subjects. Fifth, Pearson correlation analyses were performed to verify the correlations between acculturation types, acculturative stress, and social adjustment in South Korea, and multi-regression analyses and bootstrapping were performed to verify the hypotheses. The verification was carried out at significance levels of $p < 0.05$, $p < 0.01$, and $p < 0.001$.

**4. Results**

*4.1. Verification of Differences in Acculturation Types, Acculturative Stress, and Adjustment to South Korean Society According to the Characteristics of Immigrants*

4.1.1. Differences in Acculturation Types, Acculturative Stress, and Adjustment to South Korean Society According to the Ages of Immigrants

Table 3 shows the results of the analysis of the differences in acculturation types, acculturative stress, and adjustment to South Korean society according to the ages of immigrants. According to the results of the analysis, there were statistically significant differences in the factors of assimilation and marginalization among the types of acculturation according to the ages of immigrants. That is, among the types of acculturation, the ratio of assimilation was higher among immigrants in their 30s (M = 3.80) than those in their 20s (M = 3.67), and the difference was statistically significant ($p < 0.01$). The ratio of marginalization was higher among immigrants in their 20s (M = 2.00) than those in the 30s (M = 1.85), and the difference was statistically significant ($p < 0.05$).

**Table 3.** Differences in acculturation types, acculturative stress, and adjustment to South Korean society according to the ages of immigrants.

| Area | | Division | N | M | SD | t | *p* |
|---|---|---|---|---|---|---|---|
| Acculturation type | Integration | 20s | 163 | 3.93 | 0.42 | 1.431 | 0.155 |
| | | 30s | 25 | 3.87 | 0.11 | | |
| | Assimilation | 20s | 163 | 3.67 | 0.38 | −3.118 ** | 0.003 |
| | | 30s | 25 | 3.80 | 0.16 | | |
| | Separation | 20s | 163 | 1.94 | 0.40 | 1.169 | 0.244 |
| | | 30s | 25 | 1.85 | 0.12 | | |
| | Marginalization | 20s | 163 | 2.00 | 0.34 | 2.102 * | 0.037 |
| | | 30s | 25 | 1.85 | 0.07 | | |
| Acculturative stress | | 20s | 163 | 1.84 | 0.40 | −0.390 | 0.697 |
| | | 30s | 25 | 1.88 | 0.14 | | |
| Adjustment to South Korean society | | 20s | 163 | 4.09 | 0.41 | 1.088 | 0.278 |
| | | 30s | 25 | 4.00 | 0.11 | | |

* $p < 0.05$, ** $p < 0.01$.

### 4.1.2. Differences in Acculturation Types, Acculturative Stress, and Adjustment to South Korean Society According to the Levels of Education of Immigrants

Table 4 shows the results of the analysis of differences in acculturation types, acculturative stress, and adjustment to South Korean society according to the levels of education of immigrants. According to the results of the analysis, there were statistically significant differences in the factors of integration and marginalization among the types of acculturation according to the levels of education of immigrants. The ratio of integration among the types of acculturation was higher among graduate school graduates or those educated more (M = 3.97) than among college graduates (M = 3.77), and the difference was statistically significant ($p < 0.01$).

**Table 4.** Differences in acculturation types, acculturative stress, and adjustment to South Korean society according to the levels of education of immigrants.

| Area | | Division | N | M | SD | t | *p* |
|---|---|---|---|---|---|---|---|
| Acculturation type | Integration | College graduates | 47 | 3.77 | 0.39 | −3.143 ** | 0.002 |
| | | Graduate school graduates or educated more | 141 | 3.97 | 0.38 | | |
| | assimilation | College graduates | 47 | 3.70 | 0.34 | 0.411 | 0.681 |
| | | Graduate school graduates or educated more | 141 | 3.68 | 0.36 | | |
| | separation | College graduates | 47 | 2.01 | 0.33 | 1.757 | 0.081 |
| | | Graduate school graduates or educated more | 141 | 1.90 | 0.39 | | |
| | marginalization | College graduates | 47 | 1.99 | 0.29 | 0.235 | 0.814 |
| | | Graduate school graduates or educated more | 141 | 1.97 | 0.33 | | |
| Acculturative stress | | College graduates | 47 | 1.82 | 0.43 | −0.575 | 0.566 |
| | | Graduate school graduates or educated more | 141 | 1.86 | 0.36 | | |
| Adjustment to South Korean society | | College graduates | 47 | 4.04 | 0.43 | −0.761 | 0.448 |
| | | Graduate school graduates or educated more | 141 | 4.09 | 0.36 | | |

** $p < 0.01$.

### 4.1.3. Differences in Acculturation Types, Acculturative Stress, and Adjustment to South Korean Society According to the Residence Periods of Immigrants

Table 5 shows the results of the analysis of differences in acculturation types, acculturative stress, and adjustment to South Korean society according to the residence periods of immigrants. According to the results of the analysis, there were statistically significant differences in the factor separation among the types of acculturation according to residence periods of immigrants. The ratio of separation among the types of acculturation was higher among the immigrants who resided in South Korea for less than three years (M = 2.00) than among those who resided for at least three years (M = 1.87), and the difference was statistically significant ($p < 0.05$).

**Table 5.** Differences in acculturation types, acculturative stress, and adjustment to South Korean society according to the residence periods of immigrants.

| Area | | Division | N | M | SD | t | p |
|---|---|---|---|---|---|---|---|
| Acculturation type | Integration | Less than three years | 82 | 3.87 | 0.48 | −1.648 | 0.101 |
| | | At least three years | 106 | 3.96 | 0.29 | | |
| | Assimilation | Less than three years | 82 | 3.69 | 0.42 | 0.237 | 0.813 |
| | | At least three years | 106 | 3.68 | 0.30 | | |
| | Separation | Less than three years | 82 | 2.00 | 0.46 | 2.359 * | 0.019 |
| | | At least three years | 106 | 1.87 | 0.29 | | |
| | Marginalization | Less than three years | 82 | 1.99 | 0.41 | 0.537 | 0.592 |
| | | At least three years | 106 | 1.97 | 0.22 | | |
| Acculturative stress | | Less than three years | 82 | 1.86 | 0.49 | 0.386 | 0.700 |
| | | At least three years | 106 | 1.84 | 0.26 | | |
| Adjustment to South Korean society | | Less than three years | 82 | 4.05 | 0.48 | −0.762 | 0.447 |
| | | At least three years | 106 | 4.10 | 0.28 | | |

\* $p < 0.05$.

### 4.1.4. Differences in Acculturation Types, Acculturative Stress, and Adjustment to South Korean Society According to Whether Immigrants Had the Intention to Return to Their Homeland or Not

Table 6 shows the results of the analysis of differences in acculturation types, acculturative stress, and adjustment to South Korean society according to whether immigrants had the intention to return to their homeland or not. According to the results of the analysis, there were statistically significant differences in the factors of integration and separation among the types of acculturation according to whether immigrants have the intention to return to their homeland. That is, the ratio of integration among the types of acculturation was higher among the immigrants who had no intention to return to their homeland (M = 3.98) than among those who had the such intention (M = 3.71), and the difference was statistically significant ($p < 0.001$). The ratio of separation was higher among immigrants who had the intention to return to their homeland(M = 2.03) than among those who had no such intention (M = 1.90), and the difference was statistically significant ($p < 0.05$).

**Table 6.** Differences in acculturation types, acculturative stress, and adjustment to South Korean society according to whether immigrants had the intention to return to their homeland or not.

| Area | | Division | N | M | SD | t | p |
|---|---|---|---|---|---|---|---|
| Acculturation type | Integration | Yes | 42 | 3.71 | 0.35 | −4.068 *** | 0.000 |
| | | No | 146 | 3.98 | 0.38 | | |
| | Assimilation | Yes | 42 | 3.69 | 0.35 | 0.082 | 0.935 |
| | | No | 146 | 3.68 | 0.36 | | |
| | Separation | Yes | 42 | 2.03 | 0.35 | 2.050 * | 0.044 |
| | | No | 146 | 1.90 | 0.38 | | |
| | Marginalization | Yes | 42 | 2.00 | 0.30 | 0.547 | 0.585 |
| | | No | 146 | 1.97 | 0.32 | | |
| Acculturative stress | | Yes | 42 | 1.83 | 0.45 | −0.399 | 0.690 |
| | | No | 146 | 1.85 | 0.35 | | |
| Adjustment to South Korean society | | Yes | 42 | 4.05 | 0.46 | −0.447 | 0.655 |
| | | No | 146 | 4.08 | 0.36 | | |

* $p < 0.05$, *** $p < 0.001$.

### 4.2. Verification of Correlations between Variables

To find out the general trends and characteristics of the variables used in this study first before examining the influencing relationships between the acculturation types, acculturative stress, and adjustment to the South Korean society of Chinese immigrants, the analysis of descriptive statistics such as the means and standard deviations of individual variables and their subfactors and correlation analysis was conducted, and the results are as shown in Table 7. As can be seen in Table 7, the mean of the factor integration among the types of acculturation was shown to be (M) = 3.92, that of assimilation was shown to be (M) = 3.68, that of separation was shown to be (M) = 1.93, and that of marginalization was shown to be (M) = 1.98. The mean acculturative stress was shown to be (M) = 1.85, and that of adjustment to South Korean society was shown to be (M) = 4.08.

The correlations between the study variables were analyzed and according to the results, integration (r = −0.688, $p < 0.01$) and assimilation (r = −0.762, $p < 0.01$), which are subfactors of acculturation types, were negatively (−) correlated with acculturative stress and separation (r = 0.736, $p < 0.01$) and marginalization (r = 0.845, $p < 0.01$) were positively (+) correlated with acculturative stress. Integration (r = 0.649, $p < 0.01$) and assimilation (r = 0.673, $p < 0.01$), which are sub-factors of acculturation types, were positively (+) correlated with adjustment to South Korean society, and separation (r = −0.670, $p < 0.01$) and marginalization (r = −0.756, $p < 0.01$) were a negatively (−) correlation with adjustment to South Korean society. In addition, acculturative stress was shown to be negatively (−) correlated with adjustment to South Korean society (r = −0.726, $p < 0.01$).

**Table 7.** Verification of correlations between variables.

| Variable | | Acculturation Type | | | | Acculturative Stress | Adjustment to South Korean Society |
|---|---|---|---|---|---|---|---|
| | | Integration | Assimilation | Separation | Marginalization | | |
| Acculturation type | Integration | 1 | | | | | |
| | Assimilation | 0.522 ** | 1 | | | | |
| | Separation | −0.649 ** | −0.665 ** | 1 | | | |
| | Marginalization | −0.746 ** | −0.717 ** | 0.832 ** | 1 | | |
| Acculturative stress | | −0.688 ** | −0.762 ** | 0.736 ** | 0.845 ** | 1 | |
| Adjustment to South Korean society | | 0.649 ** | 0.673 ** | −0.670 ** | −0.756 ** | −0.726 ** | 1 |
| M | | 3.92 | 3.68 | 1.93 | 1.98 | 1.85 | 4.08 |
| SD | | 0.39 | 0.36 | 0.38 | 0.32 | 0.37 | 0.38 |

** $p < 0.01$.

*4.3. Hypothesis Verification*

4.3.1. Verification of Hypothesis 1

The effects of Chinese immigrants' acculturation types (integration, assimilation, separation, marginalization) on their acculturative stress were analyzed, and the results are shown in Table 8. First, whether or not there was any problem in the multicollinearity between independent variables was analyzed before analysis, and the VIF was shown to be 2.127~4.935, which was smaller than 10.0, indicating that there was no problem in the multicollinearity between independent variables. In addition, the D/W value was shown to be 2.456, which was close to 2.0, indicating that there was no correlation between the residuals.

The explanatory power of the types of acculturation of Chinese immigrants to explain their acculturative stress was shown to be = 0.774, indicating that the explanatory power was 77.4%, and F = 156.293 was shown indicating that the regression model was suitable at the significance level of $\alpha = 0.001$. Among the subfactors of acculturation types, integration ($\beta = -0.139$, $p < 0.05$) and assimilation ($\beta = -0.322$, $p < 0.001$) had significant negative ($-$) effects on acculturative stress, and marginalization ($\beta = 0.494$, $p < 0.001$) was found to have significant positive ($+$) effects on acculturative stress. The relative influence of marginalization was shown to be the largest followed by that of assimilation and integration in order of precedence. In conclusion, Hypothesis 1-1, Hypothesis 1-2, and Hypothesis 1-4 were adopted, and Hypothesis 1-3 was rejected.

**Table 8.** Effect of the types of acculturation of Chinese immigrants on their acculturative stress.

| Factor | Non-Standardized Coefficient | | Standardized Coefficient | t | p | VIF |
|---|---|---|---|---|---|---|
| | B | SE | β | | | |
| (Constant) | 2.418 | 0.423 | | 5.713 | 0.000 | |
| Integration | −0.133 | 0.051 | −0.139 | −2.617 * | 0.010 | 2.268 |
| Assimilation | −0.337 | 0.054 | −0.322 | −6.267 *** | 0.000 | 2.127 |
| Separation | 0.021 | 0.064 | 0.021 | 0.328 | 0.743 | 3.379 |
| Marginalization | 0.583 | 0.092 | 0.494 | 6.320 *** | 0.000 | 4.935 |
| Dependent variable: Acculturative stress | | | | | | |
| $R^2 = 0.774$, Adjusted $R^2 = 0.769$, F = 156.293, *** $p = 0.000$, D/W = 2.456 | | | | | | |

* $p < 0.05$, *** $p < 0.001$.

4.3.2. Verification of Hypothesis 2

The effect of acculturative stress of Chinese immigrants on their adjustment to South Korean society was analyzed, and the results are shown in Table 9. The explanatory power of the acculturative stress of Chinese immigrants to explain their adjustment to South Korean society was shown to be = 0.527, indicating that the explanatory power was 52.7%, and F = 207.159 was shown indicating that the regression model was suitable at the significance level of $\alpha = 0.001$. The acculturative stress ($\beta = -0.726$, $p < 0.001$) of Chinese immigrants was shown to have significant negative ($-$) effects on their adjustment to South Korean society. In conclusion, Hypothesis 2 was adopted.

**Table 9.** Effects of the acculturative stress of Chinese immigrants on their adjustment to South Korean society.

| Factor | Non-Standardized Coefficient | | Standardized Coefficient | t | p |
|---|---|---|---|---|---|
| | B | SE | β | | |
| (Constant) | 5.446 | 0.097 | | 56.148 | 0.000 |
| Acculturative stress | −0.741 | 0.051 | −0.726 | −14.393 *** | 0.000 |
| Dependent variable: Adjustment to South Korean society | | | | | |
| $R^2 = 0.527$, Adjusted $R^2 = 0.524$, F = 207.159, *** $p = 0.000$, D/W = 2.003 | | | | | |

*** $p < 0.001$.

### 4.3.3. Verification of Hypothesis 3

The effects of the types of acculturation (integration, assimilation, separation, marginalization) of Chinese immigrants on their adjustment to South Korean society were analyzed, and the results are shown in Table 10. First, whether or not there was any problem in the multicollinearity between independent variables was tested, and according to the results, the VIF was 2.127~4.935, which was smaller than 10.0, indicating that there was no problem in the multicollinearity between independent variables. In addition, the D/W value was shown to be 2.091, which was close to 2.0, indicating that there was no correlation between the residuals.

The explanatory power of the types of acculturation of Chinese immigrants to explain the adjustment to South Korean society was shown to be = 0.625, indicating that the explanatory power was 62.5%, and F = 76.342 was shown indicating that the regression model was suitable for the significance level of $\alpha$ = 0.001. Among the subfactors of acculturation types, integration ($\beta$ = 0.198, $p < 0.01$) and assimilation ($\beta$ = 0.268, $p < 0.001$) had significant positive (+) effects on Chinese immigrants' adjustment to South Korean society, and marginalization ($\beta$ = $-0.370$, $p < 0.001$) was shown to have significant negative ($-$) effects on adjustment to South Korean society. The relative influence of marginalization was shown to be the largest followed by that of assimilation and integration in order of precedence. In conclusion, Hypothesis 3-1, Hypothesis 3-2, and Hypothesis 3-4 were adopted, and Hypothesis 3-3 was rejected.

**Table 10.** Effects of the types of acculturation of Chinese immigrants on their adjustment to South Korean society.

| Factor | Non-Standardized Coefficient | | Standardized Coefficient | t | p | VIF |
|---|---|---|---|---|---|---|
| | B | SE | $\beta$ | | | |
| (Constant) | 3.253 | 0.555 | | 5.856 | 0.000 | |
| Integration | 0.194 | 0.067 | 0.198 | 2.898 ** | 0.004 | 2.268 |
| Assimilation | 0.286 | 0.071 | 0.268 | 4.059 *** | 0.000 | 2.127 |
| Separation | $-0.057$ | 0.084 | $-0.056$ | $-0.669$ | 0.504 | 3.379 |
| Marginalization | $-0.445$ | 0.121 | $-0.370$ | $-3.679$ *** | 0.000 | 4.935 |
| Dependent variable: adjustment to South Korean society | | | | | | |
| $R^2$ = 0.625, Adjusted $R^2$ = 0.617, F = 76.342, *** $p$ = 0.000, D/W = 2.091 | | | | | | |

** $p < 0.01$, *** $p < 0.001$.

### 4.3.4. Verification of Hypothesis 4

Bootstrapping was performed through SPSS Macro (Model 4) to verify the mediating effects of acculturative stress on the relationship between the types of acculturation of Chinese immigrants and their adjustment to South Korean society, and the results are shown in Table 11. The bootstrapping method yields a 95% confidence interval for the mediating (indirect) effect coefficient. If this confidence interval does not contain 0, it can be concluded that the mediating effect is statistically significant at the 0.05 level (Preacher & Hayes, 2004). In this study, the number of bootstrap samples was to 5000 to conduct the verification, and the upper and lower limits of the mediating effect coefficient were obtained at the 95% confidence interval. First, the integration type $\rightarrow$ acculturative stress $\rightarrow$ adjustment to the South Korean society path was analyzed and according to the result, the indirect effect coefficient was 0.3580, the lower limit value was 0.0005, and the upper limit value was 0.5400. Therefore, since 0 was not included in the confidence interval, the mediating effect was shown to be significant, and Hypothesis 4-1 was adopted. Thereafter, the assimilation type $\rightarrow$ acculturative stress $\rightarrow$ adjustment to South Korean society path was analyzed, and according to the result, the indirect effect coefficient was 0.4134, the lower limit value was $-0.01431$, and the upper limit value was 0.6154 so that 0 was included in the confidence interval, indicating that the mediating effect was not significant and Hypothesis 4-2 was rejected. Next, the separation type $\rightarrow$ acculturative stress $\rightarrow$

adjustment to South Korean society path was analyzed, and according to the result, the indirect effect coefficient was −0.3800, the lower limit value was −0.5407, and the upper limit value −0.0001, so that 0 was not included in the confidence interval, indicating that the mediating effect was significant, and Hypothesis 4-3 was accepted. Finally, the marginalization type → acculturative stress → adjustment to the South Korean society path was analyzed and according to the result, the indirect effect coefficient was −0.3111, the lower limit was −0.4585, and the upper limit was −0.0030 so that 0 was not included in the confidence interval indicating that the mediating effect was significant and Hypothesis 4-4 was accepted.

**Table 11.** Verification of the mediating effect of acculturative stress on the relationship between the types of acculturation of Chinese immigrants and their adjustment to South Korean society.

| Path | Effect | Boot SE | 95% Confidence Interval | |
| --- | --- | --- | --- | --- |
| | | | LLCI | ULCI |
| Integration → acculturative stress → social adjustment | 0.3580 | 0.1373 | 0.0005 | 0.5400 |
| Assimilation → acculturative stress → social adjustment | 0.4134 | 0.1567 | −0.0143 | 0.6154 |
| Separation → acculturative stress → social adjustment | −0.3800 | 0.1344 | −0.5407 | −0.0001 |
| Marginalization → acculturative stress → social adjustment | −0.3111 | 0.1012 | −0.4585 | −0.0030 |

## 5. Discussion and Conclusions

This study empirically analyzed the relationship among the types of acculturation of Chinese immigrants in South Korea, their acculturative stress, and their adjustment to South Korean society. A discussion centered on the results of tests of the hypotheses in this study is as follows.

First, the effects of the types of acculturation of Chinese immigrants on their acculturative stress were analyzed. According to the results, among the subfactors of acculturation type, integration ($\beta = -0.139$, $p < 0.05$) and assimilation ($\beta = -0.322$, $p < 0.001$) had significant negative (−) effects on acculturative stress, and marginalization ($\beta = 0.494$, $p < 0.001$) had significant positive (+) effects on acculturative stress. These results indicate that the greater the will and effort of immigrants in South Korea to integrate and assimilate with South Korean society, the more acculturative stress can be alleviated, whereas the more insufficient the will and effort to integrate and assimilate in South Korean society, and the stronger the marginalization to stay in the periphery, the higher the acculturative stress. Kim (2018) [59] analyzed the relationship between the acculturation type and acculturative stress of Japanese immigrants residing in Korea, and as a result, all of their integration, assimilation, separation, and marginalization significantly affected their acculturative stress. In particular, integration and assimilation were found to have positive effects on the alleviation of immigrants' acculturative stress, while separation and marginalization had negative effects. The results of that study are consistent with the results of this study, which presented significant effects for all of integration, assimilation, and marginalization, except for separation, and therefore can be regarded to support the results of this study in general.

Second, the effects of immigrants' acculturative stress on their adjustment to South Korean society were analyzed. It was found that their acculturative stress ($\beta = -0.726$, $p < 0.001$) had significant negative (−) effects on their adjustment to South Korean society. These results indicate that the degree of adjustment to South Korean society may vary depending on the management of acculturative stress. That is, if immigrants in South Korea have lower acculturative stress, their social adjustment will improve. Jin (2010) [60] empirically analyzed the relationship between the acculturative stress of immigrants of various nationalities in South Korea and their adjustment to South Korean society and the results indicated that the acculturative stress of immigrants had negative effects on their social adjustment. Those study results are consistent with the results of this study, which

showed that the acculturative stress of immigrants had significant negative (−) effects on their adjustment to South Korean society thereby supporting the results of this study.

Third, the effects of immigrants' acculturation types on their adjustment to South Korean society were analyzed. According to the results, among the subfactors of the acculturation types, integration ($\beta = 0.198$, $p < 0.01$) and assimilation ($\beta = 0.268$, $p < 0.001$) had significant positive (+) effects on adjustment to South Korean society, while marginalization ($\beta = −0.370$, $p < 0.001$) had significant negative (−) effects. Berry (2005) [32] investigated acculturation types among married immigrants and women among immigrants and reported that many immigrants experienced marginalization with low levels of the identity of their homeland culture and adjustment to the culture of migration and integration with relatively extremely high levels of cultural maintenance and adjustment to the culture of migration. He also reported that immigrants who experienced marginalization showed low levels of social adjustment, while those who experienced integration showed high levels of social adjustment. These results are generally consistent with the results of this study, thereby supporting this study.

Fourth, the mediating effect of acculturative stress on the relationship between the integration of Chinese immigrants and their adjustment to South Korean society was analyzed. As a result, it was found that the integration, separation, and marginalization of immigrants had significant indirect effects on their adjustment to South Korean society through acculturative stress. Few previous studies empirically verified the mediating effects of acculturative stress on the relationship between the integration of immigrants and their adjustment to South Korean society. However, from the results of this study, it can be inferred that integration has a positive effect on the relief of acculturative stress and that this finally has positive effects on their adjustment to South Korean society. Conversely, it can be predicted that separation and marginalization have negative effects on the relief of acculturative stress and that the aggravation of acculturative stress ultimately negatively affects their adjustment to South Korean society.

Through the results of the empirical analysis in this study, it can be seen that there are differences in the levels of acculturative stress and social adjustment experienced by foreign immigrants residing in South Korea according to their acculturation type. That is, among the acculturation types, the type of immigrant who tries to integrate and assimilate in South Korean society is not very affected by acculturative stress and adjusts to South Korean society smoothly, but the type who hovers in the periphery of South Korean society is subject to a lot of acculturative stress and has difficulties in adjustment. Therefore, for immigrants to adapt well to the culture, a variety of policy approaches are required so that immigrants can participate as independent members of Korean society. In addition, there is an urgent need for policies and programs that support immigrants to live together by actively participating in the process of social integration and forming diverse communities that help them adapt to the culture.

This study is meaningful in that it presented the acculturation types necessary for immigrants who are steadily increasing in South Korea in this era of globalization to relieve the acculturative stress they feel in an unfamiliar foreign country and adjust to South Korean society. However, this study also has the following limitations. First, since this study analyzed only 188 immigrants, there may be limitations in extending this to generalized results for all immigrants in South Korea.

Second, the immigrants are composed of various people such as married immigrant women, employed immigrants, naturalized persons, and international students, and there may be differences in acculturation types, acculturative stress, and adjustment to South Korean society according to the type of immigrant, but this study could not subdivide and deal with them in detail. In addition, there is no objective and standardized scale for measuring acculturation types, social support, acculturative stress, and adjustment to South Korean society, and there are differences in the composition of items by the researcher. Therefore, objectivity and standardization may be somewhat insufficient in the

measurement of acculturation types, acculturative stress, and adjustment to South Korean society.

Based on the limitations of this study, the following can be proposed for follow-up studies. First, this study was conducted with only 188 Chinese immigrants among foreign immigrants in South Korea. Therefore, there may be limitations in generalizing the results of this study by expanding and interpreting them for immigrants from all countries. Therefore, it will be necessary to broaden the study with a comprehensive research study that includes immigrants from other countries as well as subjects composed of only Chinese immigrants.

**Funding:** (1) This work was supported by the Hankuk University of Foreign Studies Research Fund (Of 2022). (2) This article was supported by the Ministry of Education of the Republic of Korea and National Research Foundation of Korea (NRF-2020S1A6A3A04064633).

**Institutional Review Board Statement:** Not applicable.

**Informed Consent Statement:** Informed consent was obtained from all subjects involved in the study.

**Conflicts of Interest:** The authors declare no conflict of interest.

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
