# Peer review of "Effects of Acculturation Types on Acculturative Stress and Adjustment to South Korean Society: Focusing on Chinese Immigrants"

_sustainability, doi:10.3390/su142013370_

Round 1

Reviewer 1 Report

It is an interesting study to examine the acculturation and adjustment among Chinese immigrants in Korea. While overall it seems to flow logically, please consider the following points to improve the manuscript.

1. The literature on the concept of adjustment needs to be inserted.

2. Ages or the number of residence years seem to be the possible moderators of this study. Have you considered it? What was the criteria for samples in that "all Chinses immigrants in Korea" could be quite broad?

3. The author needs to be include the legitimate logics to develop hypotheses.

4. Based on the findings, please expand theoretical and practical implications.

Author Response

Thanks for the comment.

Reviewers' opinions have been reflected as faithfully as possible.

If there are any other supplements, please comment and I will supplement them again.
thank you.

Reviewer 2 Report

This study is very interesting, up-to-date and inspiring also for researchers from other countries and continents - for example from Poland, where several million Ukrainians have found their way over the past few months, who will probably stay here longer than initially assumed.

In my opinion, work has great cognitive value. However, I have to point out a few shortcomings. Here I will only mention two of the most important. The rest are in the comments in the text. First of all, it is necessary to improve the methodological part and clarify who, when and how was tested, how the research sample was selected and how and where the questionnaire was disseminated.

Secondly, I have the impression that one of the most important characteristics of the respondents (apart from age or level of education) should be the motive / purpose of the visit. It has a great influence on the behavior of immigrants. I am aware that it cannot be supplemented now, but I suggest that you consider it in future research.

Other comments are included in the text.

Author Response

This study is very interesting, up-to-date and inspiring also for researchers from other countries and continents - for example from Poland, where several million Ukrainians have found their way over the past few months, who will probably stay here longer than initially assumed.

In my opinion, work has great cognitive value. However, I have to point out a few shortcomings. Here I will only mention two of the most important. The rest are in the comments in the text.

First of all, it is necessary to improve the methodological part and clarify who, when and how was tested, how the research sample was selected and how and where the questionnaire was disseminated.

The research method has been supplemented as follows.

3.1. Survey participants

In this study, foreign immigrants in Korea were set as the population, and among them, Chinese immigrants in their 20s and 30s, who show the highest immigration rate, were used as the sample group. The survey was conducted from the 1st week to the 4th week of April 2022 targeting the sample group, and the survey was conducted through e-mail and SNS using Google Docs. Respondents who agreed to the purpose of the questionnaire filled out the questionnaire in the form of self-filling.

Secondly, I have the impression that one of the most important characteristics of the respondents (apart from age or level of education) should be the motive / purpose of the visit. It has a great influence on the behavior of immigrants. I am aware that it cannot be supplemented now, but I suggest that you consider it in future research.

The above comments will be reflected in subsequent research.

Other comments are included in the text.

All contents of the text comments were reflected, and a PDF file was also attached. (with explanation in the comments)

Reviewer 3 Report

The authors need to first of all give the reader a better context of immigration in S.K. How many immigrants?  For the past five years? Nature of the immigrants?  Percentages?  all of this is necessary to place the research in context. the authors claim S.K. is a multicultural society.  Provide evidence of such a claim. the authors also talk about a S.K. policy in place.  tell the reader about this policy.  The definition of acculturation stress is vague and not clear. "the ability to control themselves and their surrounding environment".  We all have this issue (native born and immigrants).  Not a distinguishing feature.   Need to present more detail on the measurement procedures.  Examples of actual questions posed to the participants.

line 124 claims that "organic changes at collective and individual level"  What does this mean? 199 we don't call them "survey subjects"  the new terminology is "survey participants" or "people participating in the research." I think I know what VIF and D/W mean but readers need to be informed--and I might be wrong. 

   The authors go from generalizing from "Chinese immigrants" to "foreign immigrants".  I don't think this is possible. 

  The results section simply rehashes the data section.  Research is supposed to test theory.  There is no testing going on.  There is no explanation as to why the results were obtained.  Tell us something about the different "acculturative types" that leads them to have stress or lack of integration into S.K. society.  In short, there is no explanation as to why the results came out.  And, in the results, the authors continue to invoke literature review.  that time is passed as it was noted in the "lit review" section. 590-593 these claims are outside the results of the study.

Author Response

Thanks for the comment.

Reviewers' opinions have been reflected as faithfully as possible.

If there are any other supplements, please comment and I will supplement them again.
thank you
